# Percutaneous Interspinous Spacer in Spinal-Canal-Stenosis Treatment: Pros and Cons

**DOI:** 10.3390/medicina55070381

**Published:** 2019-07-16

**Authors:** Chiara Zini, Matteo Bellini, Salvatore Masala, Stefano Marcia

**Affiliations:** 1Dipartimento di Radiologia, Azienda USL Toscana Centro, 50012 Firenze, Italy; 2UOC NINT Neuroimmagini e Neurointerventistica, Azienda Ospedaliera Universitaria Senese, 53100 Siena, Italy; 3Diagnostica per Immagini e Radiologia Interventistica Ospedale San Giovanni Battista, 00148 Roma, Italy; 4Radiologia PO SS Trinità, ATS Sardegna ASSL Cagliari, 09121 Cagliari, Italy

**Keywords:** percutaneous interspinous devices, degenerative lumbar spinal stenosis, neurogenic intermittent claudication

## Abstract

A comprehensive description of the literature regarding interspinous process devices (IPD) mainly focused on comparison with conservative treatment and surgical decompression for the treatment of degenerative lumbar spinal stenosis. Recent meta-analysis and articles are listed in the present article in order to establish IPD pros and cons.

## 1. Introduction

Degenerative lumbar spinal stenosis (DLSS) is defined as the narrowing of the lumbar spinal canal, lateral nerve roots, and/or intervertebral neural foramina due to the progressive hypertrophy of any of the surrounding osteocartilaginous and ligamentous elements, resulting in neural and/or vascular compression and leading to neurogenic intermittent claudication (NIC) [1,2].

NIC is characterized by lower-extremity pain, relieved by setting and flexion with or without stiffness, paresthesias/weakness, and/or cramping, and it is associated with enormous economic costs, societal impairment, and health impact [1].

NIC is initially managed conservatively with medical therapies and physical therapy [2]; epidural fluoroscopic guided steroid injections are suggested to provide short-term (up to six months) pain relief in NIC [3].

When nonsurgical treatments fail, with persistent pain and/or function limitation, surgical treatment may be an option.

Surgical decompression, represented by laminectomy with/without spinal fusion, seems to improve outcomes in patients with moderate to severe symptoms of lumbar spinal stenosis, in particular when radicular pain and/or neurogenic claudication are the predominant symptoms [3,4].

DLSS is currently the leading cause of spinal surgery in patients over 65 years, and the mortality rate has been calculated to be 0.14% [5,6]; however, the postsurgical morbidity rate, represented by postoperative pain, dural tear, blood loss, infection and immobilization, is not unremarkable [7].

Because of surgery-related risks, patient comorbidities, and good results demonstrated in terms of outcomes, non-responders to conservative therapy are frequently treated with percutaneous treatments [8,9,10,11].

Interspinous process devices (IPD) are minimally invasive devices that are able to decrease facet join overload through a “shock-absorber” mechanism shifting forces to the posterior column with a reduction of discal pressure [11]; segmental enlargement of the spinal canal with the unloading of the facet joint and posterior annulus, resulting in the restoration of normal foraminal height, was reported in cadaveric studies after IPD placement [12,13].

However, complications such as dislocation and remodeling/fracture of the spinous process after IPD placement have been described in the literature and seem to be more frequent compared to patients treated with a conservative treatment [14].

There is still a lack of understanding the safety, efficacy, and cost-effectiveness of IPD [14].

Some randomized controlled trials have compared IPD to conservative treatment [15,16,17] and standard surgery [18,19,20,21,22,23,24], but all of them analyzed IPD that is no longer available in the market in the prespinoplasty era, with study analysis bias, such as follow-up period, subgroup analysis based on age and missed comorbidities, and outcome evaluation criteria, in small population samples (Table 1).

The aim of this paper was to provide complete and reliable information regarding the benefits and limits of IPD compared to conservative therapy and the surgical approach.

## 2. IPD vs. Conservative Therapy

We searched MEDLINE, EMBASE, Cochrane Library, Scopus, and LILACS for randomized and quasirandomized trials, without language or period restrictions, comparing a group in active treatment (medical/interventional) and an untreated control group (natural history); to our best knowledge, there is no direct comparison between active treatment (medical/interventional) and the untreated control group (natural history) published in the literature.

The North American Spine Society (NASS) guidelines for the diagnosis and treatment of degenerative lumbar spinal stenosis stated that there is insufficient evidence for a recommendation for or against the use of pharmacological treatment, manipulation, and the use of physical therapy or exercise as stand-alone treatments in the management of spinal stenosis [12].

NASS guidelines suggest interlaminar epidural steroid injections for short-term (two weeks to six months) pain relief in patients with neurogenic claudication or radiculopathy; however, a multiple-injection regimen, a radiographically guided transforaminal epidural steroid injection, or caudal injections are suggested to produce medium-term (3–36 months) symptom relief in patients with radiculopathy or neurogenic intermittent claudication from lumbar spinal stenosis [12].

In this scenario, IPD placement as standard procedure has been demonstrated to be more effective than conservative therapy, consisting of spinal injections, anti-inflammatory and analgesic drugs, and physical therapy (Level 1) [15,16,17]; however, no evaluation of pain and overall functional status has been reported, and no cost analysis comparing IPD with conservative treatment has been performed [15,16,17].

The Zurich Claudication Questionnaire (ZCQ) was administered in two series, concluding that patients in the IPD group improved significantly more than the conservative-therapy control group (*p* < 0.0001) [15,17]; in particular, the baseline ZCQ score was significantly improved at all postoperative periods. Anderson et al. reported that the IPD baseline ZCQ score of 50.4 dropped to 23.1 at the two-year follow up, while there was no significant improvement in the control-group ZCQ score at any follow-up interval [15].

Quality of life, evaluated with SF-36, was significantly improved in the IPD group compared with the conservative control group, especially in physical activity (*p* < 0.001) [16].

However, complications were most frequent in the IPD group (11 in 93 patients vs. six in 81 for controls) and the treatment failure rate, defined by surgical conversion (laminectomy), was higher in the IPD group (six patients out of 93) compared to the conservative-therapy control group (24 patients out of 81; RR 0.22 (0.09, 0.51)) [14]; subgroup analysis that evaluates patients with listhesis had similar results [15].

## 3. IPD vs. Surgery

NASS guidelines suggest decompressive surgery in patients with moderate to severe symptoms because of lumbar spinal stenosis; however, interventional treatment may be considered for patients with moderate symptoms of lumbar spinal stenosis [12].

Decompression alone is suggested for patients with leg predominant symptoms without instability, while decompression with spinal fusion is considered in patients with spondylolisthesis, even those over 75 years old [12,25].

Seven trials have compared the outcomes of different forms of IPD with surgical spinal decompression [18,19,20,21,22,23,24]; however, the evaluated devices in those studies are mini-invasive (not completely percutaneously placed) and no longer available on the market. The comparison between IPD and surgical procedure was also inhomogeneous (minimally invasive surgery (two trials), decompression and arthrodesis (two trials), and decompression alone (three trials)) [18,19,20,21,22,23].

Recent meta-analysis reported no significant difference between the IPD group and surgical group in term of pain evaluation using VAS score (*p* = 0.09). Although pooled VAS leg-pain data favor the IPD group—it is important to underline that the indication of IPD placement is NIC—the difference was not clinically significant [14].

Overall functional status presented no significant difference between the two groups [14]; however, high heterogeneity has been registered because the studies used different evaluation methods, such as the Oswestry Disability Index (ODI) [18,19], the Modified Roland Disability Questionnaire [22], or nonspecific functional scales [23].

The two-year ZCQ follow-up showed no significant difference between the IPD group and surgical group in term of symptom severity, physical function, and patient satisfaction [19,20,21,22,23,24].

The overall described reoperation rate was higher in the IPD group except in the Azzazzi et al. series [18] because of dislocation of the device and erosion/fracture of the spinous process after IPD placement [14,18,19,20,21,22,23]; data from Meyer et al. underline that the high re-intervention rate in the IPD group (29% vs. 6% after 12 months, and 26% vs. 8% after 24 months) was mostly due to “lack of success” instead of complications as described in the surgical group, concluding that a selection of patients is mandatory for the indication of IPD placement [24].

Recent meta-analysis focused on complications revealed no statistical difference in terms of complications between the IPD group and the surgical group in those series that reported it [14,19,22,23,24]; it is important to underline that the use of cement augmentation of the posterior vertebral arch (spinoplasty) in order to reduce the risk of spinal process fracture and IPD displacement has not been reported in any of the trials comparing IPD versus surgery for the treatment of DLSS (Figure 1) [25,26].

In terms of cost effectiveness, the IPD has been claimed to be far below 50% when compared to a surgical procedure, and the cost of the device was significantly higher than surgical decompression [27]. However, in the cost analysis reported by Lonne et al., the IPD incremental cost compared with minimally invasive decompression was €2832 (95% confidence interval: 1886–3778), whereas the incremental health gain was 0.11 quality-adjusted life-years (95% confidence interval: −0.01–0.23) [21].

## 4. IPD: Pros and Cons

IPD are mini-invasive devices placed using fluoroscopic guidance between the adjacent spinous process at the DLSS level, sparing anatomical structures of the spinal canal in order to relieve nerve compression using distractive force applied by the device with subsequent height restoration [27].

IPD can be implanted under local anesthesia plus sedation in a daycare surgery, and it has been demonstrated to be faster and lead to less blood loss compared to a surgical procedure because part of the procedure is performed percutaneously [24]; recently, a brand-new IPD with completely percutaneous placement (Lobster, TechlaMed, Firenze, Italy) has been proposed, but no trials have been performed using this particular device.

There are many different IPD designs using different materials, such as allograft, titanium, and polyetherethrketone (PEEK), but based on the mechanical characteristics, IPD can be divided in two groups: interspinous distraction devices (IDD) that act to separate adjacent spinous processes, and the more recent interspinous stabilizers (ISS) that are affixed statically (for example, X-Stop (Medtronic), Wallis (Zimmer Biomet), and Superion (Vertiflex)) or dynamically (for example, the Coflex (Paradigm Spine) and DIAM (Medtronic)) adjacent spinous processes [27,28] (Figure 2 and Figure 3).

From the first “metal plug” in 1950 to the brand-new Lobster (Lobster, TechlaMed, Firenze, Italy), the fate of IPD has changed; in the beginning, many procedures were performed with a lack of patient selection and a procedure learning curve until the recent ban of almost all IPD from the market.

IPD may be regarded as a less invasive alternative to open decompression in patients with DLSS, avoiding typical surgical complications such leaks and epidural hematomas [24] (Figure 4); however, a high complication rate has been described in almost all trials comparing surgical approaches with IPD [19,20,21,22,23,24].

It is important to clarify that IPD treatment failure can be related to “real” complications such as device displacement and spine process erosion/rupture, or to “lack of success” with mandatory surgical revision [24].

Device displacement may be related to posterior ligament impairment because of the surgical time to place the IPD; mistaken patient selection, especially because of anatomical difficulties, could also be responsible for IPD displacement.

On the other hand, it has been demonstrated that the erosion/rupture of the spinous process during or after IPD placement can be avoided using prophylactic cement augmentation of the posterior vertebral arch (spinoplasty) [24,25]; the published data in the literature demonstrated that no restenosis occurred at the 3–12 month follow-up, and, in the case of a second IPD implant at the same level after spinoplasty, obtained a resolution of symptoms with no further bone remodeling [25].

In a case of “lack of success”, Meyer et al. suggested that it is almost related to patient selection and biomechanical reasons [24]; in fact, Moojen et al. suggested that the cross-sectional area of the spinal canal may be inferior after IPD placement if compared to surgical laminectomy, leading to earlier symptom relapse in those patients with severe DLSS treated with IPD [22].

IPD cost effectiveness is still debated, mainly because of a different evaluation in terms of complication rate and subsequent reoperation costs [21,27].

## 5. Conclusions

IPD are diverse mini-invasive devices placed with fluoroscopic guidance under local anesthesia between the spinal processes at the DLSS level in order to obtain nerve decompression.

It has been demonstrated to be more effective than a conservative treatment for DLSS; treatment failure appeared to be significantly lower in the IPD group, while complications seemed to be more frequent for the implant group compared to the conservative treatment [15,16,17].

Low quality evidence indicated that outcomes regarding pain, functional status and quality of life are similar comparing IPD with surgical procedures; however, treatment failure was significantly higher in IPD group compared to decompressive surgery because of complication as dislocation of the device and erosion/fracture of the spinous process-that could be avoided with spinoplasty-or “lack of success” almost related to patient selection [28,29,30].

Cost-effectiveness of IPD is still debate.

A prospective randomized study to evaluate the efficacy of pure percutaneous IPD + preventive spinoplasy versus spinal laminectomy with long (>24 months) term follow-up is highly desirable.

## Figures and Tables

**Figure 1 medicina-55-00381-f001:**
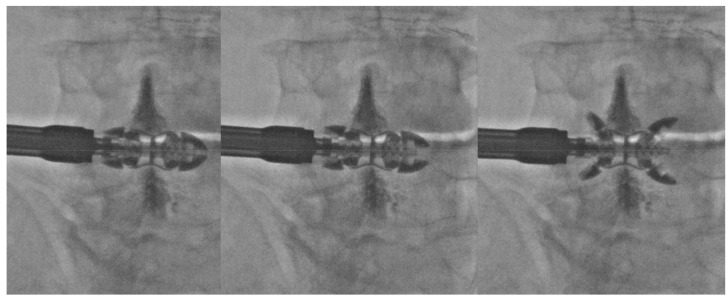
Anteroposterior fluoroscopic images of progressive opening of Lobster device (Lobster, TechlaMed, Firenze, Italy) in order to prevent lateral dislocation; preventive spinoplasty performed to prevent spinal process remodeling/rupture.

**Figure 2 medicina-55-00381-f002:**
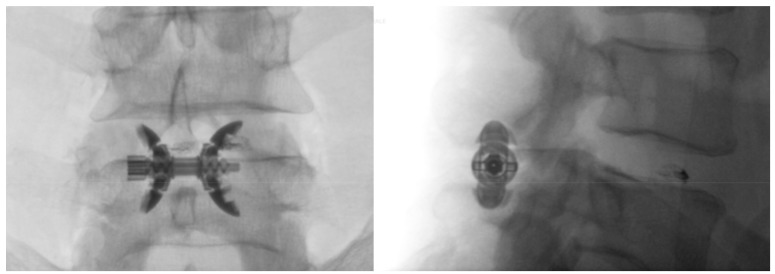
Anteroposterior and lateral fluoroscopic images of Lobster device (Lobster, TechlaMed, Firenze, Italy) device correctly placed at the level of L4–L5; the polyetherethrketone (PEEK) part of the device is radiolucent. Previous discography at the level of the L4–L5 disc.

**Figure 3 medicina-55-00381-f003:**
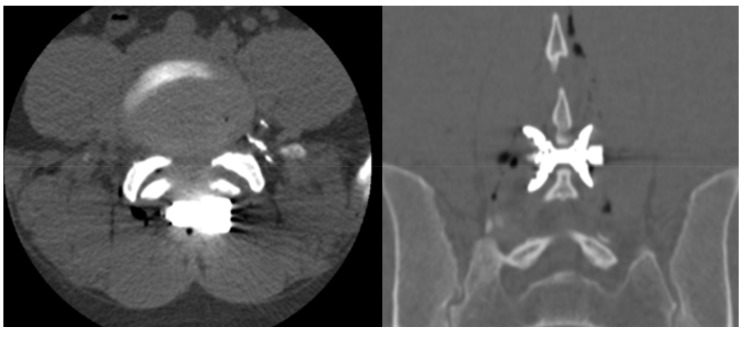
Axial computed tomography image (CTI) and coronal multiplanar reconstruction demonstrated the correct placement of Lobster device (Lobster, TechlaMed, Firenze, Italy).

**Figure 4 medicina-55-00381-f004:**
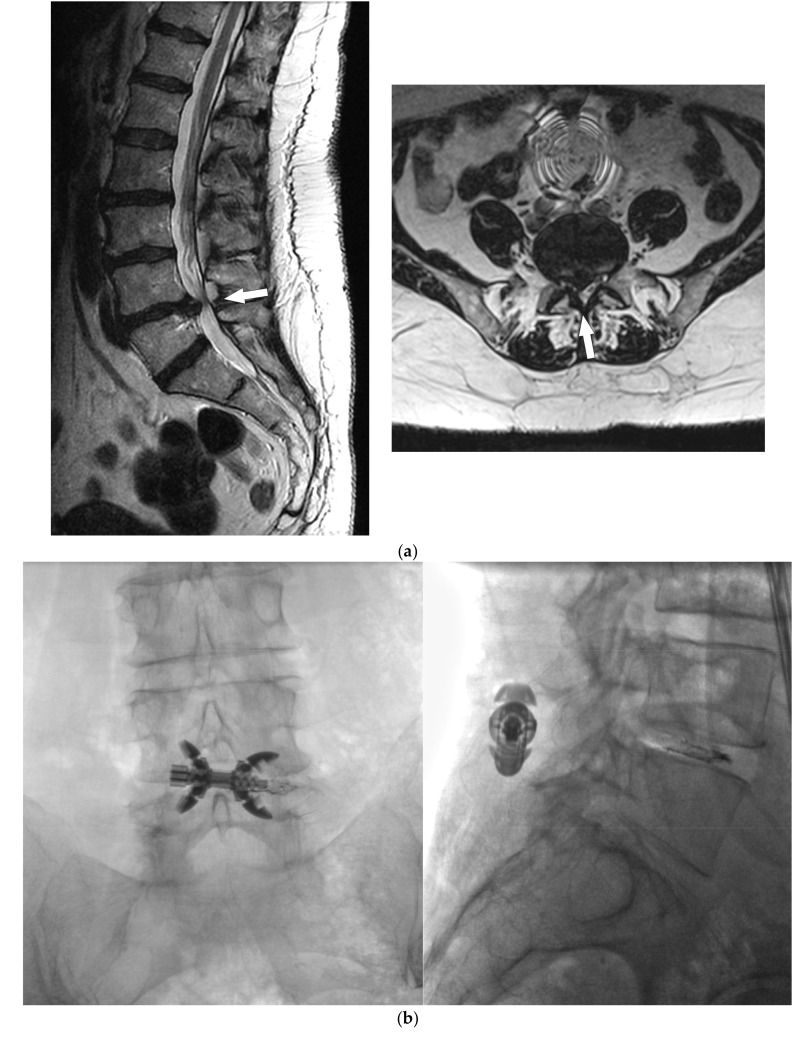
Fifty-five-year-old male with NIC because of DLSS at the L4–L5 level. (**a**) Axial and sagittal T2-weighed MR images demonstrated right subarticular L4–L5 disc herniation and spinal canal stenosis (arrow). (**b**) Anteroposterior and lateral fluoroscopic images showed correct placement of 10 mm Lobster (TechlaMed, Firenze, Italy) device made with titanium and PEEK at the L4–L5 level; previous chemodiscolisys was performed at the level of the L4–L5 disc for treatment of disc herniation. (**c**) One-week follow-up MR axial and sagittal T2-weighed images demonstrated enlargement of spinal canal with normal foraminal height and reduction of herniation (arrows).

**Table 1 medicina-55-00381-t001:** Description of included studies.

Study	Study Design	Population	Outcomes
Anderson/Hsu/Zucherman, 2005 [15,16,17]	RCT, multicenter, IPD (X-Stop) vs. nonsurgical treatment	-Mean age 70 (IPD), 69.1 (control) -Clinical or radiographic DLSS confirmation -One or two levels affected -Able to sit 50 min and walk > 50 ft -Nonoperative treatment > 6 months	-ZCQ [15,17] -SF-36 [16] -Patient satisfaction
Azzazzi, 2010 [18]	RCT, single center, IPD (X-Stop) vs. surgery (decompression and arthrodesis)	-Mean age 57 (IPD), 56.3 (control) -DLSS + grade I listhesis -One or two affected levels -Leg pain > back pain -Nonoperative treatment > 3 months	-VAS back pain -VAS leg pain -ODI
Davis, 2013 [19]	RCT, multicenter, IPD (Coflex) vs. surgery (decompression and arthrodesis)	-Mean age 62.1 (IPD), 64.1 (control) -NIC and radiographic confirmation of DLSS -One or two affected levels -VAS back pain > 50 -ODI > 20/50	-ZCQ -VAS back pain -VAS leg pain -ODI -SF-12
Stromkvist, 2013 [23]	RCT, multicenter, IPD (X-Stop) vs. surgery (decompression)	-Mean age 67 (IPD), 71 (control) -NIC and DLSS confirmation on MRI - >six month symptoms -One or two affected levels -up to Grade 1 listhesis	-ZCQ -VAS back pain -VAS leg pain -SF-36
Lønne, 2015, [20,21]	RCT, multicenter, IPD (X-Stop) vs. surgery (minimally invasive decompression)	-Mean age 67 (IPD), 67 (control) -NIC and DLSS confirmation on MRI -One or two affected levels -up to Grade I listhesis	-ZCQ -Numerical pain scale -ODI -EQ-5D -QALY
Moojen, 2015 [22]	RCT, multicenter, IPD (coflex) vs. surgery (decompression)	-Median age 66 (IPD), 64 (control) -NIC and DLSS confirmation on MRI -One or two affected levels	-ZCQ -VAS back pain -VAS leg pain -McGill pain questionnaire -RMQ -SF-36 -HADS -Shuttle walking test

IPD = interspinous process device, RCT = randomized controlled trial, ZCQ = Zurich Claudication Questionnaire, SF-36 = Medical Outcomes Study 36-Item Short-Form Health Survey, DLSS = degenerative lumbar spine stenosis, VAS = visual analogue scale, ODI = Oswestry Disability Inventory, NIC= Neurogenic intermittent claudication, MRI = magnetic resonance image, EQ-5D = EuroQol five-dimension scale, SF-12 = Medical Outcomes Study 12-Item Short-Form Health Survey.

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
