# Peer review of "Percutaneous Interspinous Spacer in Spinal-Canal-Stenosis Treatment: Pros and Cons"

_medicina, 2019, doi:10.3390/medicina55070381_

Reviewer 1 Report

Title: Percutaneous Interspinous Spacer in Spinal Canal Stenosis Treatment: Pros and Cons

Reviewer Comments

Summary

This manuscript specifically discussed the pros and cons of Interspinous process devices (IPD) by comparing them with conservative treatment and surgical decompression technique.  Although a systematic review of the data is presented and discussed, very critical information related to the performance of IPD devices currently existed in the market and improvements needed for the long-term survival of these devices is missing. Additionally, major English language editing is required. Therefore, I will recommend this article for publication with major changes.

Specific Comments:

Introduction:

1)      Page 1, Lines 26: it should be “in NIC patients”

2)      Page 2, Lines 45: seemed to be more frequent – Please provide numbers.

IPD vs Conservative Therapy

1)      Page 2, Lines 55-56: It is a strong statement. Did you conducted any literature search to come to this conclusion? If so, please provide details about database search, article filtering etc.

2)      Page 2, Lines 71-73: Please provide numbers for ZCQ study.

3)      Page 2, Lines 74-75: Please provide numbers for the study.

IPD vs Surgery

1)      Page 3, Lines 93-96: Please provide numbers for the study.

2)      Page 3, Lines 119-122: Please provide current cost range for both the techniques.

3)      Page 4, Line 137:  Zimmer should be “Zimmer Biomet”

A  table summarizing the studies and respective findings should be included.

Author Response

We thank you for your review and comments.

We change line 26 and line 137, as you suggested.

We provided the numbers from the died studies at line 71-73,74-75 and 93-96; the numbers from the studies cited at line 45 are discussed extensively in section 2 (IPD vs Surgery)

Line 55-56  We searched MEDLINE, EMBASE, Cochrane Library, Scopus, and LILACS for randomized and quasi-randomized trials, without language or period restrictions, comparing active treatment (medical/interventional) and untreated control group (natural history); we were not able to find any strong data that is the reason why we wrote that kind of sentence.

Line 119-122 The cost of devices and procedures is highly variable; we do prefer to add the numbers from the cost analysis made by Lonne et al.

We added the table resuming the cited studies.

Reviewer 2 Report

General impression

In this article, the authors well summarized benefits and limits of the percutaneous interspinous spacers compared to conservative therapy and surgical treatment.  I believe the information in this study must be valuable for the physicians to manage lumbar spinal canal stenosis.  Therefore, I think this manuscript is appropriate for publication.  However, I have a couple of minor requests to be revised as stated below.  After these have been resolved, I will judge this manuscript can be accepted and published by the medicina journal.

 *I hope correction parts will be shown in red color in the revised manuscript.

 1. Figure 4

  Resolution of pictures of (b) in the Figure 4 is extremely intolerable.  I think they should be replaced with clearer ones.

 2. IPDs and IPDS

  In this article, IPDs (“s” is a small letter) and IPDS (“S” is a capital letter; line 91) are mixed.  They should be unified either.

 3. IPD group and IPDs group

  In this article, IPD (singular) and IPDs (plural; line 110) are mixed.  They should be unified either.

 Author Response

We thank you for your review and comments.

We made the changes you asked in the text and we have correct as possible the figure 4 as you suggest.

Round  2

Reviewer 1 Report

Please include the sentence related to literature search in the manuscript.

We searched MEDLINE, EMBASE, Cochrane Library, Scopus, and LILACS for randomized and quasi-randomized trials, without language or period restrictions, comparing active treatment (medical/interventional) and untreated control group (natural history);

Author Response

Dear Reviewer1,

we thank you for your comment; we add the sentence you recomened in the text.

Thanks for your time

The Authors